**PLOS** NEGLECTED TROPICAL DISEASES

# The role of connectivity on malaria dynamics across areas with contrasting control coverage in the Peruvian Amazon

**Gabriel Carrasco-Escobar**[1,2]*, **Diego Villa**[1], **Antony Barja**[1], **Rachel Lowe**[3,4], **Alejandro Llanos-Cuentas**[5], **Tarik Benmarhnia**[6]

**1** Herbert Wertheim School of Public Health and Human Longevity Science, University of California San Diego, La Jolla, California, United States of America, **2** Health Innovation Laboratory, Institute of Tropical Medicine "Alexander von Humboldt", Universidad Peruana Cayetano Heredia, Lima, Peru, **3** Centre for Mathematical Modelling of Infectious Diseases, London School of Hygiene & Tropical Medicine, London, United Kingdom, **4** ICREA Barcelona Supercomputing Center—Centro Nacional de Supercomputación (BSC-CNS), Life & Medical Sciences, Barcelona, Spain, **5** Instituto de Medicina Tropical Alexander von Humboldt, Universidad Peruana Cayetano Heredia, Lima, Peru, **6** Scripps Institution of Oceanography, University of California, San Diego, California, United States of America

* gabriel.carrasco@upch.pe

**Data Availability Statement:** The data analyzed in this study is subject to the following licenses/ restrictions: Dataset are owned by a third-party

## Abstract

Network analysis may improve the understanding of malaria epidemiology in rural areas of the Amazon region by explicitly representing the relationships between villages as a proxy for human population mobility. This study tests a comprehensive set of connectivity metrics and their relationship with malaria incidence across villages with contrasting PAMAFRO (a malaria control initiative) coverage levels in the Loreto department of Peru using data from the passive case detection reports from the Peruvian Ministry of Health between 2011 and 2018 at the village level. A total of 24 centrality metrics were computed and tested on 1608 nodes (i.e., villages/cities). Based on its consistency and stability, the betweenness centrality type outperformed other metrics. No appreciable differences in the distributions of malaria incidence were found when using different weights, including population, deforested area, Euclidian distance, or travel time. Overall, villages in the top quintile of centrality have a higher malaria incidence in comparison with villages in the bottom quintile of centrality (Mean Difference in cases per 1000 population; *P. vivax* = 165.78 and *P. falciparum* = 76.14). The mean difference between villages at the top and bottom centrality quintiles increases as PAMAFRO coverage increases for both *P. vivax* (Tier 1 = 155.36; Tier 2 = 176.22; Tier 3 = 326.08) and *P. falciparum* (Tier 1 = 48.11; Tier 2 = 95.16; Tier 3 = 139.07). The findings of this study support the shift in current malaria control strategies from targeting specific locations based on malaria metrics to strategies based on connectivity neighborhoods that include influential connected villages.

## Author summary

In our study, we explored how the connections between villages in the Amazon region can help us better understand the spread of malaria. By examining how people move

organization and will be available upon request to the Peruvian Ministry of Health. Requests to access these datasets should be directed to transparencia@minsa.gob.pe. Code used in this study is available at https://github.com/healthinnovation/network-malaria.

**Funding:** The author(s) received no specific funding for this work.

**Competing interests:** The authors have declared that no competing interests exist.

between these villages, we identified key locations that play a significant role in the spread of the disease. We used data from 2011 to 2018, collected from the Peruvian Ministry of Health, focusing on the Loreto department. Our analysis involved 1608 villages, where we computed and tested 24 different connectivity metrics to see which one best predicted malaria cases. We found that the "betweenness centrality" metric, which measures how often a village serves as a bridge between others, was the most reliable predictor of malaria incidence. Interestingly, we discovered that villages with higher connectivity tended to have more malaria cases. This trend was even more pronounced in areas with stronger malaria control efforts. Our findings suggest that current malaria control strategies could be improved by focusing not just on individual villages with high malaria rates but also on those that are well-connected to others. This approach could lead to more effective interventions and a better understanding of how diseases like malaria spread in rural areas.

## 1. Introduction

The Peruvian Amazon is experiencing epidemiological changes in malaria transmission as a result of landscape modifications, climatic factors, malaria control interventions, and anthropogenic drivers. Regionally, malaria epidemiology is dominated by *P. vivax* (80%), with remaining 20% of cases attributed to *P. falciparum* [1]. Currently, the Loreto department accounts for an estimated 90% of all malaria cases reported in Peru [2]. In this area, between 2006 and 2010, an intense malaria control program, PAMAFRO, (Project for Malaria Control in Andean Border Areas) was undertaken, supported by the Global Fund [3]. This program was successful and effective, resulting in a sharp reduction in malaria, with cases reaching their lowest number (22,909) in 2011[4]. However, since 2011 this trend has been reversed, with a peak of 61,108 malaria cases reported in 2014 [2]. Re-emergence factors such as asymptomatic reservoirs [5,6], meteorological conditions [7], and changes in the mosquito population [8–10] have been previously studied in this area. However, evidence of other factors such as human population mobility (HPM) and, as a result, the connectivity between villages with contrasting malaria transmission remains scarce.

The transit and return of people from locations with contrasting endemicity levels must be addressed to achieve malaria elimination [11]. This flow, also referred to as connectivity, influences the endemicity level in the system (group of villages/cities) and jeopardizes control interventions that focus on targeted villages as isolated from (not connected to) other locations. This human population flow between two areas influences, to some degree, the malaria endemicity and risk in both locations (origin and destination). Under the World Health Organization (WHO) malaria elimination framework, human mobility and connectivity are key parts of the malariogenic potential, defined as the likelihood that an imported infection establishes local malaria transmission due to characteristics of the host, the parasite, the vector, and the ecosystem [12]. Therefore, understanding the role of human mobility in malaria transmission is critical for informing control efforts. By identifying how mobility patterns contribute to the spread of malaria, we can develop more effective intervention strategies that target highly connected regions, ultimately aiding in the goal of malaria elimination.

The relationships between two or more ecological entities (i.e., villages) are often analyzed as networks [13–15]. Different properties can help capturing the level of connectivity between such entities and can be measured at the entities (nodes), the links, or at the overall level, and were recently used in the analysis of infectious diseases. Buckee et al. analyzed the malaria parasite population structure from serological networks [16]. Tatem et al. estimated the role of

international population movements on *P. falciparum* malaria elimination strategies [17]. Pindolia et al. further analyzed regional connectivity and the mobility of different demographic groups in in East Africa and showed that demographically-stratified HPM and malaria movement estimates using network analysis can provide quantitative evidence to inform the design of more efficient malaria interventions [18]. And Huang et al. expanded this analysis to understand the global malaria connectivity through air travel and showed that both malaria-free areas and other endemic regions are strongly connected, particularly in Africa and Southeast Asia [19].

However, no agreement has yet been reached on which network property best captures how HPM affects malaria epidemiology, particularly in areas such as Latin America, which is the region with the most rapid urban growth rate in the world [20,21]. Furthermore, the current projections of population growth in the Amazon region involve dramatic changes in natural landscapes but also in human behaviors such as HPM. In consequence, in this study we aimed to investigate how human population mobility, quantified through various connectivity metrics, influences malaria transmission by different levels of PAMAFRO coverage in the Loreto department of Peru between 2011 to 2018. We proposed to incorporate land use and land cover (LULC) changes to reflect the expansion of villages and cities nested in watersheds that reflect microcircuits of mobility. Taken together, refined metrics of connectivity between villages have the potential to better inform malaria control efforts. In this study, we used data from the passive case detection (PCD) reports from the Peruvian Ministry of Health (MoH) between 2011 and 2018 at the village level to test a comprehensive set of connectivity metrics, including population and environmental (deforestation) weights, and their relationship with malaria incidence across villages with contrasting baseline malaria transmission and PAMAFRO coverage levels in the Loreto department of Peru.

## 2. Methods

### 2.1. Study design

This is an observational ecological study that tests the relationship between connectivity metrics and the malaria incidence in the Loreto department of Peru. Connectivity metrics were derived from the combination of multiple centrality types (i.e., betweenness, strength, eigen, and closeness) and weights such as masses (i.e., population and deforested area) and costs (i.e., distance and travel time). The relative importance of the nodes has been analyzed as a driver for malaria incidence in the area using ten-year (2011–2018) records of the PCD data from the MoH at village/city (node) level. This relationship was further stratified across villages with contrasting baseline malaria transmission and PAMAFRO coverage levels.

### 2.2. Study area

The Loreto department, located in the northeast of Peru, covers 28.7% of the national territory and a total population of 883,510. The political-administrative organization of Loreto is divided into 8 provinces, 53 districts and 31 watersheds (Fig 1). Iquitos is the capital city and the most densely populated with 510,000 inhabitants (the 7th most populated city in Peru). Most inhabitants (69.6%) live in urban areas, 32.2% live in poverty, and 7% live in extreme poverty. Only 39.6% of households have access to basic public services (water, sanitation, electricity, and telephone) [22]. Most common economic activities are based on agriculture, fishing, and mining [22]. In 2016, Loreto had a total of 521 health care facilities, with a ratio of 1,086 inhabitants per healthcare personnel [22] and important transportation and monetary barriers [23,24] to quality health care access. The tropical climate in this area ranges on average

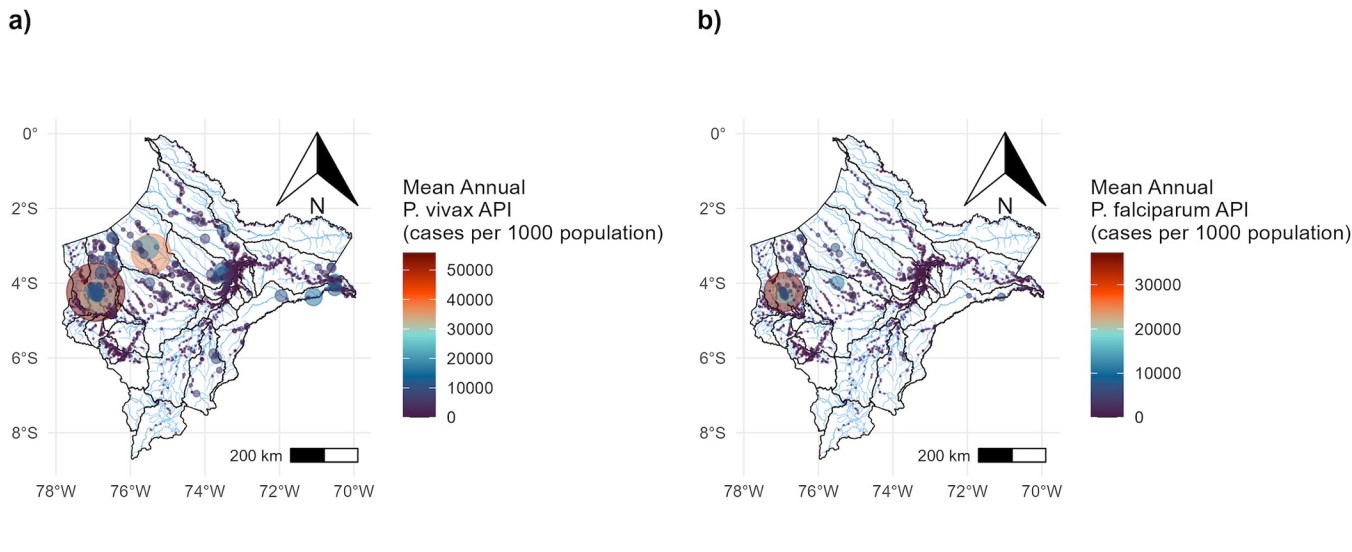

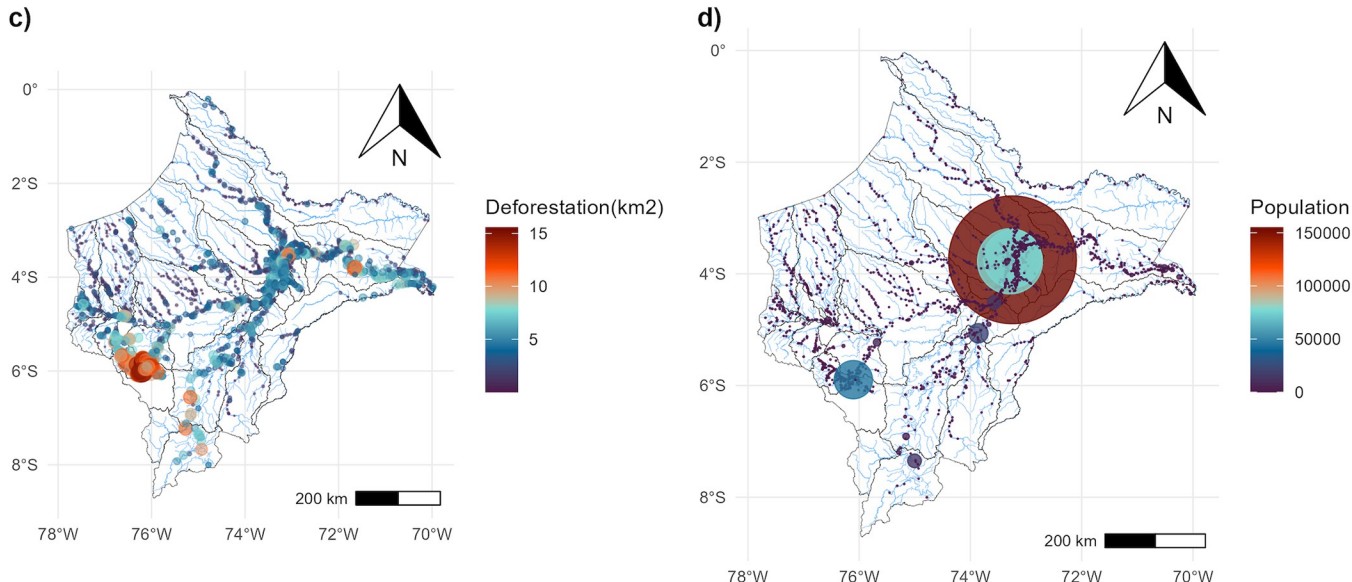

**Fig 1. Study area and hydro-basins in the Loreto department in the Peruvian Amazon.** Each point represents the location of villages and the color and size represent their A) Mean Annual *P. vivax* API, B) Mean Annual *P. falciparum* API, C) Deforested area, and D) Population size. Maps were produced using R v.4.1 (R Development Core Team, R Foundation for Statistical Computing, Australia) based on public geographic data extracted from Peruvian Open Data Portal (https://github.com/healthinnovation/network-malaria/blob/main/data/processed/03_geometries_district.zip) under Open Data Commons (ODC-By) v1.0 (http://opendefinition.org/licenses/odc-by/).

from 17˚C (between June and July) to 36˚C (between December and March) with a rainy season between December and March.

## 2.3. Data sources

**2.3.1. Malaria passive case detection data.** Malaria is a notifiable disease by the Peruvian MoH and the registry of individual-level data started in 2009 [1]. These data are available in both electronic and hardcopy format for the dominant malaria species *P. vivax* and *P.*

*falciparum*. In Peru, malaria diagnosis relies primarily on microscopic inspection of thick and thin blood smears in health facilities. The presence of asexual and sexual stages of *Plasmodium* species is determined after examining 100 high-powered fields [25]. All positive cases are immediately treated according to national guidelines from MoH [26]: chloroquine (CQ) for 3 days and primaquine (PQ) for 7 days in confirmed *P. vivax* malaria infections, and mefloquine (MQ) for 2 days and artesunate (AS) for 3 days in confirmed *P. falciparum* infections. For this study, georeferenced data were obtained at the village/city level from 2011 to 2018 for each month and were collapsed for the entire study period. GPS coordinates of the centroids and population size are provided for each village by the MoH. Malaria endemicity level was computed by the (2011–2018) Annual Parasite Index (API) as the total number of cases per 1000 population.

**2.3.2. Deforestation and watershed data.**   The Hansen collection [27], a high-spatial resolution (1 arc-second, approximately 30 meters) dataset of yearly forest coverage loss, was used to extract village-level (within 5 km radius from the centroids of villages/cities) mean deforested area (2011–2018, Km$^2$). The Hansen collection defines forest loss as a stand-replacement disturbance, or a change from a forest to non-forest state, using the year 2000 as reference and bands 3, 4, 5, and 7 of Landsat 7 cloud-free image composites. Hansen collection data were gathered and processed in Google Earth Engine [28], a cloud-based platform for planetary-scale geospatial analysis (https://earthengine.google.com). The watershed boundaries were obtained from the Peruvian National Authority of Water (ANA by its Spanish acronym). ANA provides a division, codification, and systematization of watersheds using two international standard methodologies, the Pfafstetter coding system [29] and a Digital Elevation Model (DEM) such as NASA's SRTM of 30 meters spatial resolution. The final product is a map to a scale of 1:100,000 cm.

**2.3.3. Distance and travel time estimation.**   The computation of the distance from each village to all villages analyzed in the entire department of Loreto was performed by calculating the Euclidean distance using the R Statistical Software (v4.2.2; R Core Team 2021). Euclidean distance assumes direct travel paths and may not accurately represent actual travel routes, which can be influenced by road networks and geographical barriers. Despite this limitation, Euclidean distance provides a simplified and computationally efficient method to analyze connectivity. This Euclidean distance is defined as the shortest straight line that exists between two points without considering the type of existing surface. For its calculation the following formula was used:

$$d_{(x_{(i,j)}, y_{(i,j)})} = \sqrt{(x_j - x_i)^2 + (y_j - y_i)^2}$$

Where $(x_i, y_i)$ are the coordinates of the origin and $(x_j, y_j)$ are the coordinates of the destination.

The estimation of travel time was conducted in R Statistical Software (v4.2.2; R Core Team 2021) using the rgee package [30] that bridges R to the Google Earth Engine (GEE) API [28]. We followed travel time estimation procedures described in previous literature [23,31]. To summarize the method, information about land coverage, road infrastructure, and river network was used to create a 30-m resolution grid surface. The speed assigned for each category of land cover was obtained from elsewhere [31] and the Ministry of Transportation provided the speed for the road infrastructure. Accuracy of speed restriction estimates should be validated with mayor precision at local level in future studies to better represent study settings. A friction surface was constructed where each pixel contained the cost (time) to move through the area encompassed in the pixel. Then, a cumulative cost function was applied (least-cost-

path algorithm) that examined all potential paths iteratively, and the time-weighted cost was then minimized to calculate the minimum travel time between villages.

## 2.4. Network analysis

From the Euclidean distance and travel time calculations, we obtained two origin-destination datasets of all possible connections between villages in the study area. After data cleaning and harmonization (S1 Methods), we used these interactions (links) to construct graph class objects in R for visualization and calculation of centrality metrics for each community within each of the watersheds (Fig 2). Network processing and visualization was performed using R Statistical Software (v4.2.2; R Core Team 2021). The standardized mean and standard deviation of the computed metrics were estimated overall and by watershed.

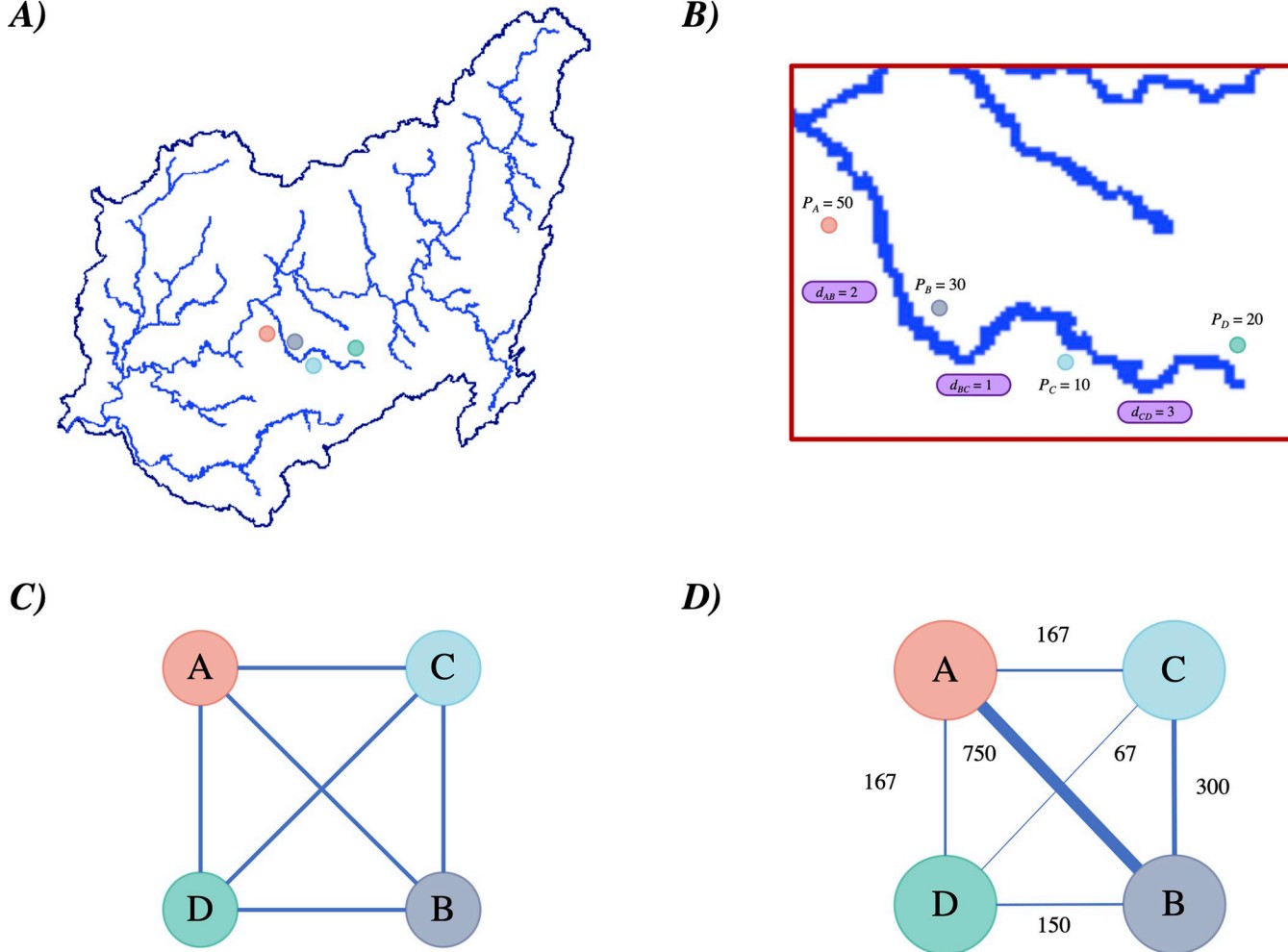

**Fig 2. Connectivity and centrality estimation workflow.** Synthetic example of all the steps to compute the centrality metrics that comprises A) the geolocation of the river network and villages in each watershed area. B) Estimate the cost of displacement between villages (i.e. distance and travel time). C) Construction of an undirected and unweighted network based on the connections between villages in the same watershed. Finally, D) testing gravity model weights for the links in the network. Weights were computed using multiple masses (i.e. population and deforested areas). Edges widths relative to weights. Maps were produced using R v.4.1 (R Development Core Team, R Foundation for Statistical Computing, Australia) based on public geographic data extracted from Peruvian Open Data Portal (https://github.com/healthinnovation/network-malaria/blob/main/data/processed/03_geometries_district.zip) under Open Data Commons (ODC-By) v1.0 (http://opendefinition.org/licenses/odc-by/).

**2.4.1. Network processing.** In this study, the 1,608 communities were considered as nodes and the 73,944 possible connections as the edges. The origin-destination dataset containing the connections was formatted as an edge list, as each row represents an edge. The distances and travel times computed for each connection were assigned iteratively as edge weights, which may represent the strength or weakness of the connection between nodes. Additionally, we constructed four different versions of weights based on the gravity model by combining population and average annual forest loss as masses with Euclidean distance and travel time as cost proxies. Finally, we scaled the weights to range from 0 to 1 within each watershed.

Having the nodes and edges with their different weights, we used the tidygraph package in R [32] to create the graphs for each watershed. Since the connections between the villages are unique and do not have a directional sense, the resulting graphs are undirected graphs. Next, we calculated the centrality metrics. For this study, the strength, closeness, betweenness, and eigenvector centralities were assessed (S2 Methods). The calculations of these centralities were made considering the weight of the edges. Each type of centrality works with a different interpretation of the weights. These measures were chosen to provide a comprehensive view of how well-connected regions are within the network, each highlighting different facets of influence and accessibility. This metrics evaluated crucial aspects such as: 1) the number of direct connections to a node, providing insights into immediate contacts, 2) the average shortest path from a node to all others, reflecting accessibility and potential for rapid dissemination, 3) nodes that serve as bridges within the network, crucial for understanding pathways of pathogen transmission, and 4) the influence of a node's connections, highlighting nodes connected to other well-connected nodes. As mentioned earlier, the weights represent the relative importance of the connection between two nodes. In the case of strength and eigenvector centrality indicators, the weight is interpreted as connection strength and, therefore, the higher the value of the weight the greater the connection between the nodes. On the other hand, in the case of closeness and betweenness centrality indicators, the weight is interpreted as connection weakness and, therefore, the higher the value of the weight the lower the connection between the nodes. For these reasons, since distance and travel time measure how far apart the communities are, we use the inverse of these measures for the calculations of the strength and eigenvector centralities. On the other hand, since the weights based on the gravity model represent the "attraction" between communities, we used the inverses of these weights for the calculations of the closeness and betweenness centralities.

We thus obtained six different versions of the centrality indicators depending on the weight used: 1) Euclidean distance, 2) Gravity model with distance and population, 3) Gravity model with distance and forest loss, 4) Travel time, 5) Gravity model with travel time and population, and 6) Gravity model with travel time and forest loss. Within every watershed, we calculated all the versions of the centrality indicators for each village and then scaled them from 0 to 1 using tidygraph. The correlation between all centrality metrics across all villages were computed using a Pearson correlation and dendrograms to cluster centrality metrics were based on a hierarchical cluster analysis using a complete linkage method.

**2.4.2. Network visualization.** Network visualizations were constructed using the ggraph package in R [33]. Two types of visualizations were used for all versions of the centrality metrics. The first consisted of plotting all nodes and edges and distinguishing them by watershed using colors, plotting the opacity of the edges as a function of the weight value (higher weight, less opaque), and setting the node size as a function of the centrality index value (higher centrality, larger size). The second type of visualization consisted of plotting the networks in different facets for each watershed and following the same settings for edge opacity and node size as in the first type of visualization. For both visualizations, two node layout algorithms were

tested: the Kamada-Kawai algorithm and the Stress majorization algorithm. Both algorithms emerge from the same optimization problem, however, the second one uses a more global approximation technique to the problem, resulting in improvements in run time and stability of the resulting node layout [34]. The algorithm used for the final visualizations was chosen by visual inspection of how well the nodes were arranged for our data.

## 2.5. Stratified analysis

Further explorations of the relationship between connectivity metrics and malaria incidence were conducted by stratifying the data across levels of intervention coverage of the PAMAFRO project (2006–2010). Four control activities were recorded per district and year including: i) strengthening of malaria diagnosis, ii) training and supervision of community health workers, iii) community-based larval source management (LSM), and iv) distribution of long-lasting insecticidal nets (LLINs). A more detailed description of control activities carried out during the 2006–2010 intensified malaria control period (PAMAFRO) are found elsewhere [3]. The PAMAFRO intervention coverage was computed as the proportion of intervention-years (maximum of 4 interventions multiplied by 5 years; 20 intervention-years) conducted in each district and assigned to all of that district's villages (code repository: https://github.com/healthinnovation/network-malaria).

## 3. Results

The main findings of this study indicate a significant relationship between connectivity metrics and malaria incidence. High malaria incidence was linked to highly connected villages, especially near Iquitos and in deforested areas. Betweenness centrality consistently demonstrated the highest stability among 24 computed metrics, highlighting its importance in understanding malaria transmission. Villages in the top quintile of betweenness centrality had significantly higher malaria incidence compared to those in the bottom quintile, with this relationship being more pronounced in areas with higher PAMAFRO coverage. In the following section we will describe the baseline characteristics of the villages and centrality estimates to provide further context to our findings, demonstrating how centrality patterns are related to higher malaria rates.

### 3.1. Baseline characteristics of villages

In total, data from 1,608 nodes (villages/cities) nested within 31 watersheds in the Loreto department were analyzed after data cleaning (S1 Fig). The total population in the selected villages was 15,045 inhabitants, and 232,252 *P. vivax* and 60,512 *P. falciparum* cases were georeferenced to the village level during the 2011–2018 period. The average number of villages in the selected watersheds is 51 ranging from 8 to 202 villages (Table 1). Most populated villages/cities are located on the banks of rivers, mainly close to Iquitos city; in contrast, most highly deforested areas are located on the south-west side of the study area (Fig 1).

Important spatial heterogeneity was observed for both *P. vivax* and *P. falciparum* cases. The highest Annual Parasite Index (API) was observed in the watersheds of Pastaza, Tigre, Yavari, and Napo (Fig 1). Historically, these watersheds showed contrasting trajectories. Pastaza and Tigre watersheds showed a rapid increase in the malaria incidence, on the other hand, Napo and Yavari watersheds, despite its high malaria endemicity level, were stable during the 2011–2018 period (Fig 3). This scattered spatial location of villages and cities is reflected in the contrasting distributions of distance and travel time between village/city dyads (S2 Fig). Both are positively skewed; however, a larger kurtosis is present in the travel time distribution in comparison to Euclidian distance. This pattern is consistent across all the 31 watersheds (S3 Fig).

**Table 1. Descriptive demographical, epidemiological, and environmental characteristics (2011–2018) in all villages nested in 31 watersheds in the Loreto department, Peru.**

| Watershed Name | Number of villages | Total number of cases | | Deforestation | |
|---|---|---|---|---|---|
| | | *P. falciparum* | *P. vivax* | Mean | sd |
| Cuenca Carhuapanas | 32 | 10 | 664 | 5.41099 | 3.35717 |
| Cuenca Itaya | 69 | 2415 | 21452 | 4.75289 | 2.53186 |
| Cuenca Manití | 14 | 327 | 1228 | 2.23778 | 1.79954 |
| Cuenca Morona | 46 | 595 | 2488 | 1.29575 | 0.88265 |
| Cuenca Nanay | 91 | 10276 | 47046 | 4.43810 | 3.29485 |
| Cuenca Napo | 169 | 6333 | 28177 | 3.15514 | 2.35080 |
| Cuenca Paranapura | 86 | 285 | 5073 | 9.75672 | 5.39148 |
| Cuenca Pastaza | 95 | 16825 | 34343 | 1.51598 | 1.24127 |
| Cuenca Potro | 8 | 15 | 310 | 1.35012 | 0.73314 |
| Cuenca Putumayo | 42 | 232 | 1037 | 1.07531 | 1.03147 |
| Cuenca Tahuayo | 17 | 10 | 213 | 4.36052 | 2.01849 |
| Cuenca Tapiche | 35 | 1365 | 4956 | 2.42216 | 2.55880 |
| Cuenca Tigre | 75 | 11830 | 33253 | 2.33620 | 1.84907 |
| Cuenca Yavari | 38 | 2815 | 10177 | 2.92976 | 2.09816 |
| Intercuenca 4977 | 202 | 1820 | 16051 | 4.27788 | 3.04429 |
| Intercuenca 49791 | 12 | 7 | 107 | 4.31276 | 2.35457 |
| Intercuenca 49793 | 40 | 125 | 1441 | 5.53734 | 3.08079 |
| Intercuenca 49795 | 8 | 4 | 34 | 4.88276 | 2.69110 |
| Intercuenca 49797 | 49 | 82 | 1580 | 4.65196 | 1.88203 |
| Intercuenca 49799 | 35 | 6 | 155 | 4.14853 | 1.97210 |
| Intercuenca 49871 | 11 | 16 | 418 | 6.73127 | 3.70163 |
| Intercuenca 49873 | 10 | 3 | 32 | 4.31457 | 2.41013 |
| Intercuenca 49877 | 42 | 130 | 989 | 3.49134 | 2.32782 |
| Intercuenca 49911 | 19 | 18 | 254 | 3.32678 | 1.99509 |
| Intercuenca 49913 | 72 | 42 | 618 | 3.95505 | 2.86444 |
| Intercuenca 49915 | 9 | 1 | 26 | 5.73575 | 2.71946 |
| Intercuenca Bajo Huallga | 62 | 136 | 500 | 7.78687 | 5.19647 |
| Intercuenca Bajo Marañón | 48 | 329 | 8600 | 3.89483 | 2.19509 |
| Intercuenca Medio Bajo Huallaga | 36 | 67 | 336 | 10.66703 | 6.19642 |
| Intercuenca Medio Bajo Marañón | 124 | 4383 | 10620 | 2.10576 | 2.06783 |
| Intercuenca Medio Marañón | 12 | 10 | 74 | 3.27035 | 2.35675 |

## 3.2. Centrality estimation

Multiple connectivity metrics were computed from the combination of centrality types and weights. A consensus graph was constructed to represent the network between villages located in the same watershed using the multiple iterations of centrality metrics. An example using betweenness centrality with weights based on a gravity model that includes Euclidian distance and population is shown in Fig 4. Due to the high density of villages and links, a version divided by watersheds is presented in S4 Fig. The densest networks are Intercuenca 4977, Napo, Medio Bajo Marañon, Pastaza, and Nanay.

A total of 24 centrality metrics were computed. Overall summary statistics are shown in Table 2 and watershed-specific statistics are shown in S1 Table. Metrics with the highest standardized mean and variability are those that use distance and travel time as weights. On the other hand, metrics with the lowest standardized mean and variability are those that use a gravity model based on distance and population or travel time and population. Betweenness

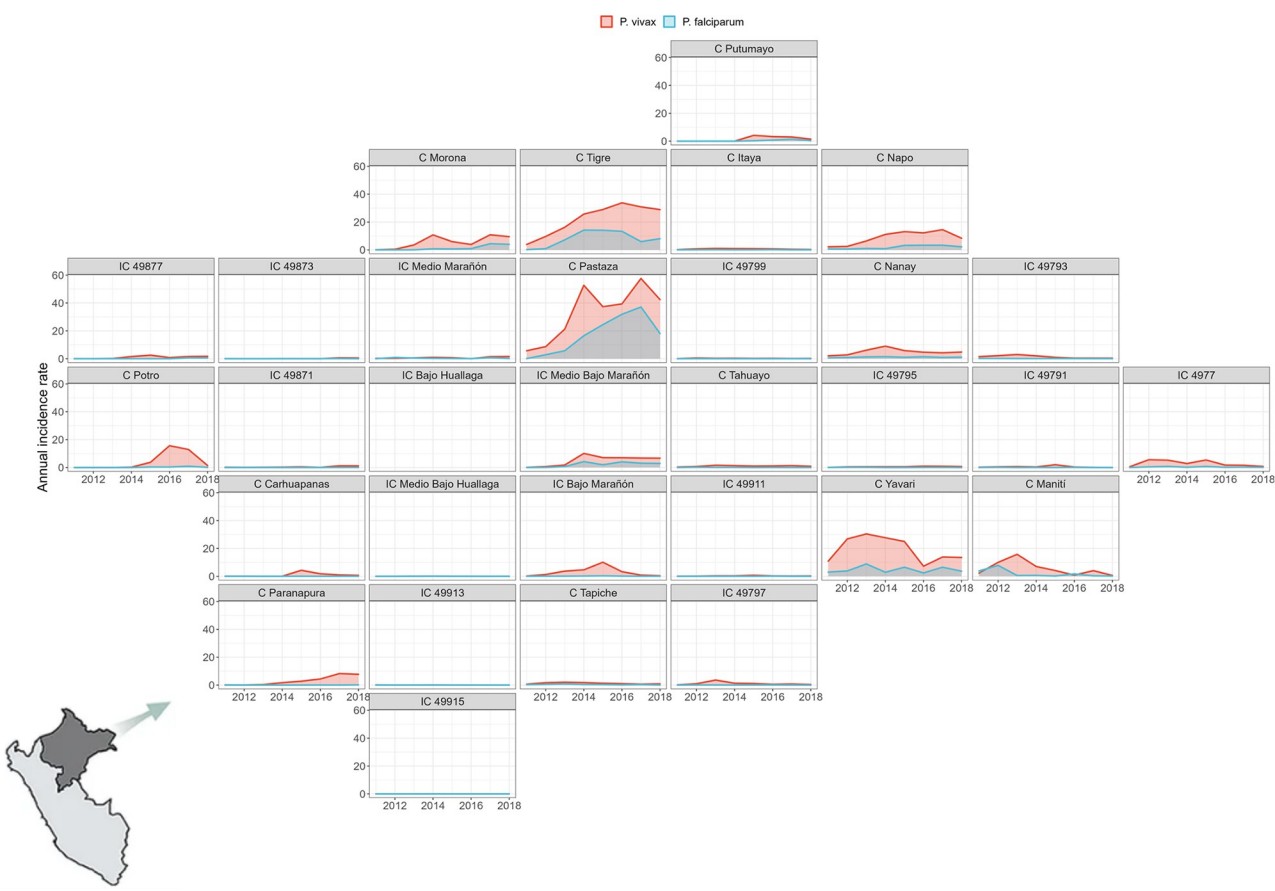

**Fig 3. Annual malaria incidence rates variation by parasite species.** Annual malaria incidence rates variation due to *P. vivax* (red) and *P. falciparum* (light blue) in 31 watersheds of Loreto department between 2011 and 2018. Maps were produced using R v.4.1 (R Development Core Team, R Foundation for Statistical Computing, Australia) based on public geographic data extracted from Peruvian Open Data Portal (https://github.com/healthinnovation/network-malaria/blob/main/data/processed/03_geometries_district.zip) under Open Data Commons (ODC-By) v1.0 (http://opendefinition.org/licenses/odc-by/).

centrality is consistently the metric with the lowest standardized mean and variability across all weights (Table 2) and these trends are consistent across watersheds (S1 Table). Strong correlation patterns were observed between centrality metrics (Fig 5). Betweenness centrality is the metric that showed the most consistent clustering pattern in the hierarchical clustering analysis (S5 Fig).

## 3.3. Relationship of centrality with malaria incidence

Betweenness centrality indicators using multiple versions of gravity models as weights were selected for further analyses based on the criteria described above. This metric was used to define categories of centrality (low vs. high quintiles) to test their relationship with malaria incidence. Overall, villages in the top quintile of centrality have a higher malaria incidence in comparison with villages in the bottom quintile of centrality (Mean Difference [MD] in cases per 1,000 population; *P. vivax* = 165.78 and *P. falciparum* = 76.14) (S6 Fig). When stratifying by levels of PAMAFRO coverage, the mean difference between villages at the top and bottom centrality quintiles increase as PAMAFRO coverage increase for both *P. vivax* (Tier 1 = 155.36; Tier 2 = 176.22; Tier 3 = 326.08) and *P. falciparum* (Tier 1 = 48.11; Tier 2 = 95.16; Tier 3 = 139.07). Overall distributions are comparable across calculations of weights (i.e.,

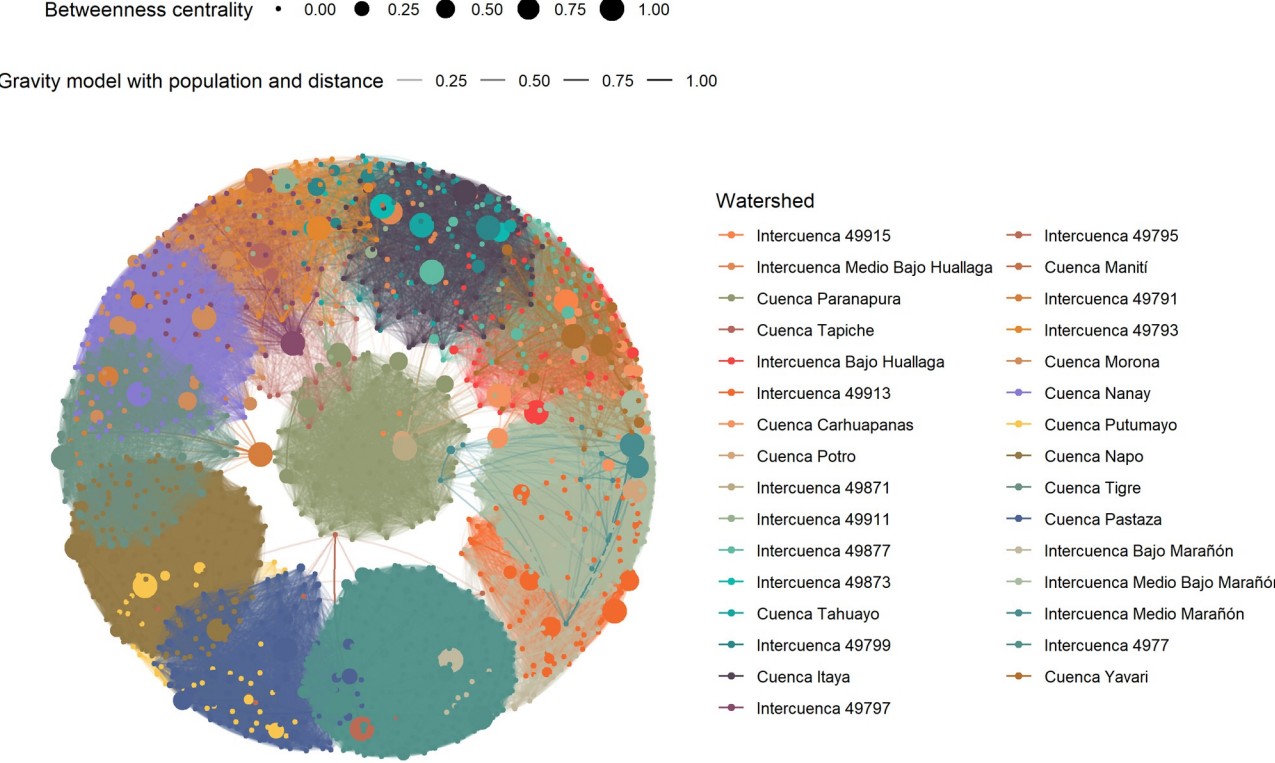

**Fig 4. Consensus graph of the network of villages in the Loreto department in the Peruvian Amazon.**

combinations of distance/travel time and population/deforestation); however, the dose-response pattern following the PAMAFRO coverage is consistent across all combinations of weight calculations for *P. falciparum* in contrast to *P. vivax* where the pattern is more noticeable when using deforestation instead of population (Fig 6).

## 4. Discussion

Our findings indicate that betweenness centrality, among the various connectivity metrics, consistently and stably predicts malaria incidence. This result held true across different weighting methods, including population, deforested area, Euclidean distance, and travel time. Specifically, regions with higher betweenness centrality consistently exhibited higher malaria incidence rates. Further analysis revealed that this pattern is particularly pronounced in areas with extensive malaria control activities from the PAMAFRO project, which initially targeted regions with high baseline malaria transmission. This underscores the need to reconsider current malaria control strategies. Instead of focusing solely on locations based on their malaria metrics, our findings support a shift towards strategies that target connectivity neighborhoods, incorporating influential connected villages that facilitate the flow of parasites and hosts. Understanding how villages and cities are connected and how these connections influence the transmission of pathogens is of paramount importance for global public health. This study contributes to the limited literature on human mobility and its impact on malaria in rural areas of the Amazon region by investigating this relationship in the Peruvian Amazon. We utilized a comprehensive set of connectivity metrics and malaria incidence records at a granular spatial resolution. In summary, our study highlights the critical role of human connectivity in malaria transmission in the Loreto department of Peru. By prioritizing highly connected

**Table 2. Descriptive statistics of centrality metrics in all villages in the Loreto department, Peru.**

| Centrality | Mean (sd) |
|---|---|
| *Distance as weight* | |
| Strength | 0.68 (0.26) |
| Closeness | 0.65 (0.28) |
| Betweenness | 0.14 (0.24) |
| Eigenvector | 0.64 (0.28) |
| *Distance and population based gravity model weight* | |
| Strength | 0.11 (0.18) |
| Closeness | 0.19 (0.21) |
| Betweenness | 0.03 (0.15) |
| Eigenvector | 0.12 (0.18) |
| *Distance and forest loss based gravity model weight* | |
| Strength | 0.27 (0.26) |
| Closeness | 0.39 (0.29) |
| Betweenness | 0.10 (0.23) |
| Eigenvector | 0.26 (0.26) |
| *Travel time as weight* | |
| Strength | 0.62 (0.28) |
| Closeness | 0.64 (0.28) |
| Betweenness | 0.12 (0.23) |
| Eigenvector | 0.60 (0.30) |
| *Travel time and population based gravity model weight* | |
| Strength | 0.09 (0.17) |
| Closeness | 0.18 (0.20) |
| Betweenness | 0.02 (0.14) |
| Eigenvector | 0.10 (0.18) |
| *Travel time and forest loss based gravity model weight* | |
| Strength | 0.22 (0.26) |
| Closeness | 0.33 (0.28) |
| Betweenness | 0.12 (0.23) |
| Eigenvector | 0.21 (0.26) |

regions, we can enhance the effectiveness of malaria control efforts and move closer to the goal of malaria elimination.

The main mechanism of malaria transmission reestablishment is the importation of parasites from HPM, and this plays a major role in elimination scenarios. However, it is also meaningful at the micro-geographical level, after malaria control interventions occur and malaria endemicity is expected to be low with only remaining *P. vivax* hypnozoite reservoirs. In these scenarios, interrupting importation pathways may greatly improve the effectiveness of current malaria control efforts [11,35]. However, capturing individual HPM information requires intensive use of resources [36–39]. The findings of this study showed that the use of connectivity metrics between villages contributes to an improved understanding of these complex dynamics in rural areas that are highly connected through river networks.

Importantly, we found that in areas with the greatest malaria endemicity and coverage of PAMAFRO control activities, the influence of connectivity was more prominent. These findings challenge previous literature that highlighted a greater importance of HPM in low-transmission and close-to-elimination settings than in moderate- and high-transmission settings [12,17,35,40,41]. In fact, less attention was put into the role of HPM and connectivity in high

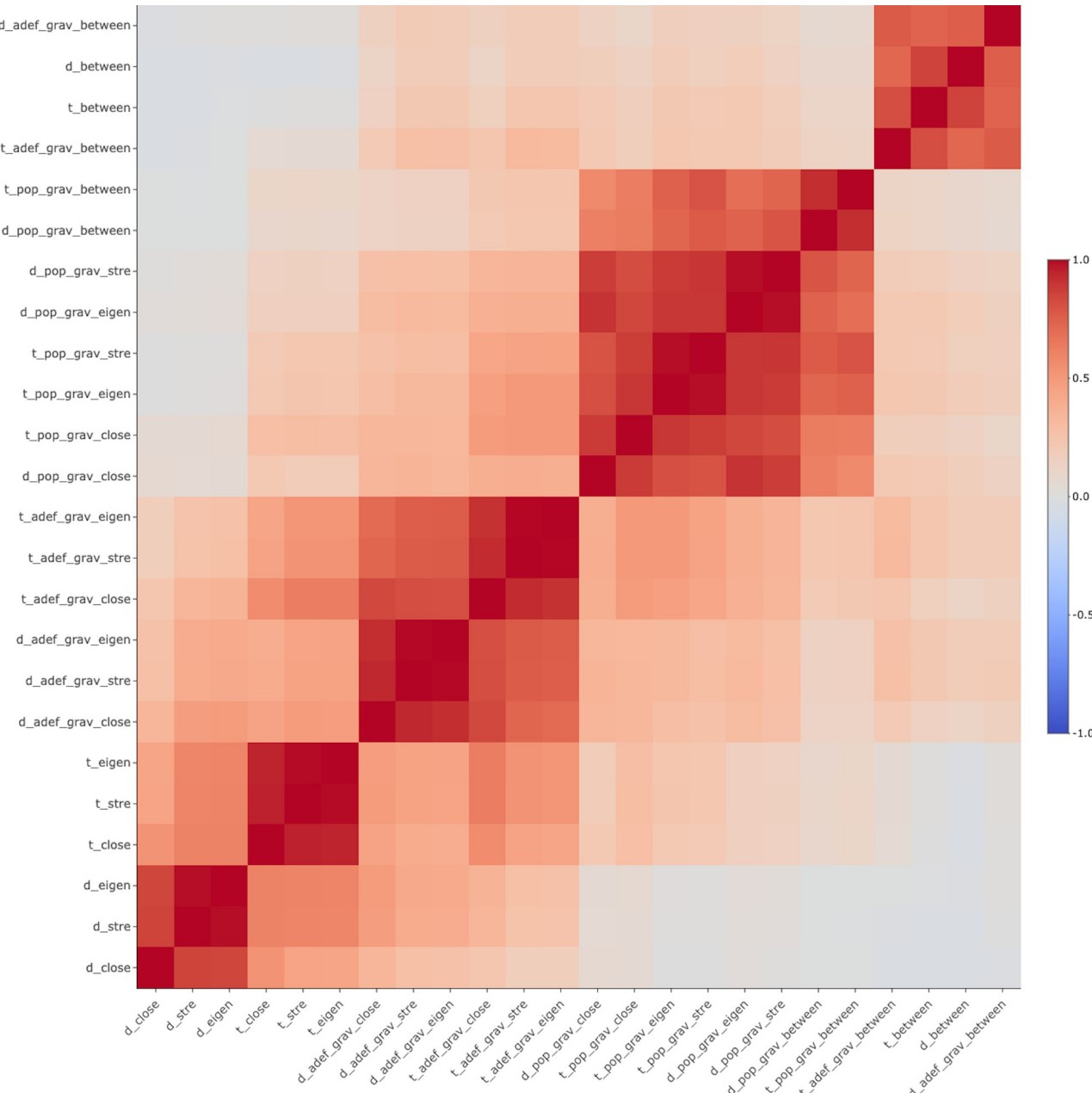

**Fig 5. Correlation of centrality metrics of villages in the Loreto department in the Peruvian Amazon.** Abbreviations: Mass (population [pop], deforested area [adef], and none), cost (distance [d], travel time [t]), and centrality type (betweenness [between], strength [stre], eigen [eigen], and closeness [close]).

malaria transmission settings [42,43]. Interestingly, areas with high vectorial capacity [8] and parasite genomic diversity [44] are areas with intense HPM in the Peruvian Amazon. However, these findings are consistent with an emergent body of evidence showing the role HPM in high- to moderate-transmission settings [36,45–50].

PAMAFRO intervention was temporally associated with a decrease in malaria transmission in Loreto Department [1]. However, from 2012 to the present, both *P. vivax* and *P. falciparum* malaria cases have rapidly increased, highlighting the fragile nature of these gains [1,51]. The

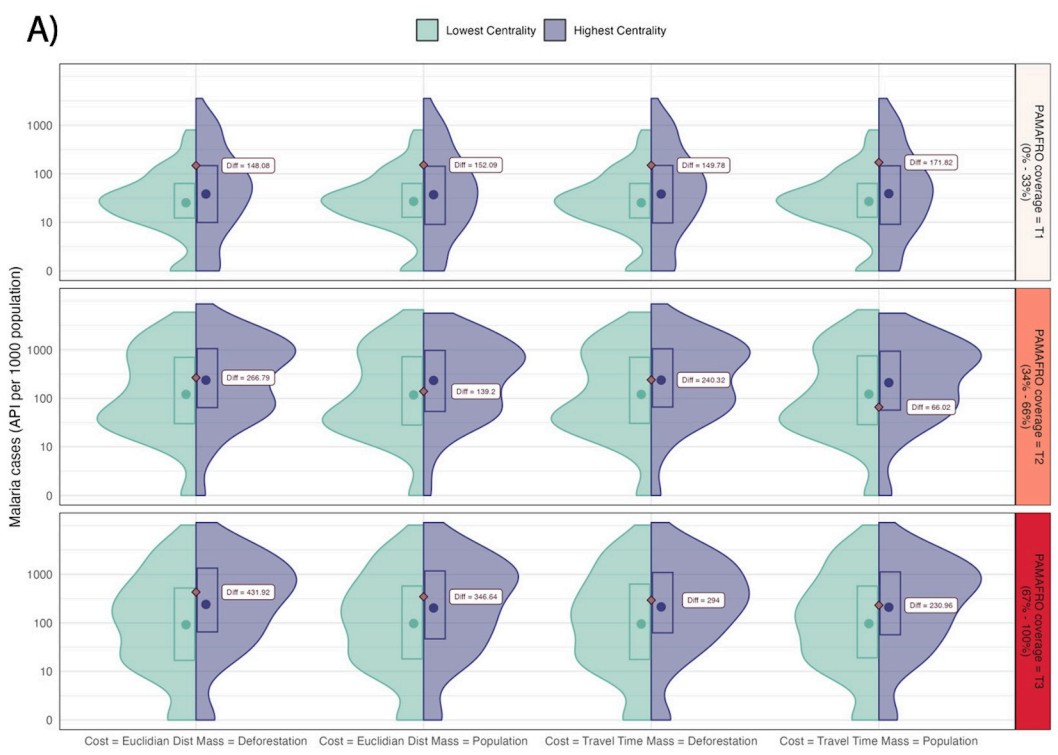

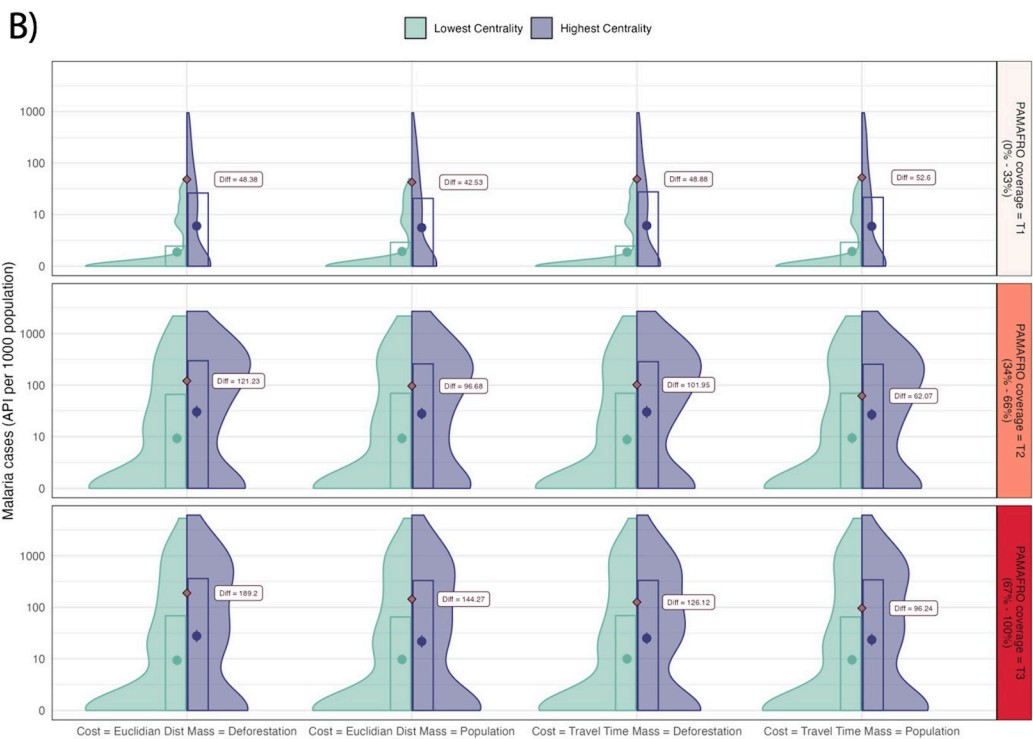

**Fig 6. Distribution of Total Annual Parasite Index (API) per 1000 population (2011–2018) across high and low centrality villages stratified by levels of PAMAFRO intervention coverage in the Loreto department in the Peruvian Amazon.** A) For *Plasmodium vivax* and B) for *P. falciparum*. Mean difference between groups are represented as diamonds in each panel. T1 = Tier 1 (Low coverage; Low baseline endemicity), T2 = Tier 2 (Moderate coverage; Moderate baseline endemicity), and T3 = Tier 3 (High coverage; High baseline endemicity).

interruption of PAMAFRO led to a resurgence of malaria in Loreto, with precipitation and actual evapotranspiration having a more significant effect on *P. falciparum* compared to *P. vivax* [7]. This implicates that malaria control policies should include the re-establishment of continuous vector control measures, community health education, and robust surveillance systems [52–54]. Therefore, Incorporating climate data into predictive models can help in anticipating and mitigating outbreaks [51].

In the Amazon region, as cities grow, HPM intensifies, and as a result so does the probability of malaria importation. The magnitude of attraction in these areas is affected by the size of the cities as a proxy for the number of services and level of commercial activity in place. In addition, these anthropogenic environmental changes impact infectious disease dynamics [55]. Increased human population and environmental modification influence biological communities, including *Anopheles* mosquitoes, particularly those with some degree of competence to transmit *Plasmodium* sp. that circulate in the Amazon region [56,57]. In this study, centrality metrics computed using population size and deforested area showed comparable performance. These similarities may be leveraged in scenarios with weak vital registration statistics such as rural areas, areas under conflict, or forced displaced populations [58–61] since the collection of deforested area could be conducted using remote sensing tools in comparison to the intense effort involved in a population census.

Nodes with high degree centrality, particularly those with high indegree, are crucial importation points and potential outbreak hubs, while nodes with high outdegree are significant in exporting malaria to other areas. Evaluating directed graphs with differing indegree and outdegree values for each node allows us to identify and target these critical points within the network. Nodes with high centrality values were pivotal during malaria epidemics, indicating that these nodes play a significant role in the spread of the disease [62]. In addition, Woolhouse *et al.*[63] and Smith *et al.*[64] emphasize the importance of tailored interventions based on the population biology of multihost pathogens and the basic reproductive number for malaria. This finding suggests that intensified surveillance and targeted control measures at these nodes could substantially reduce transmission rates.

In this study, connectivity and centrality measures were assessed in relationship to land coverage change using network analysis at the village level in Peru, and their effect on malaria transmission was estimated. This evidence contributes to the understanding of the role of HPM in malaria transmission in rural areas, and secondary, provides information to optimize the distribution of services or the configuration of networks to reduce the overall flow of malaria infections between cities and villages.

We acknowledge some limitations of this study. First, 221 villages (12%) were excluded from the analysis in the data cleaning process due to missing data of the masses and costs for the weight calculations (S1 Fig). These exclusions may alter the estimations; however, data was missing completely at random (MCAR), reducing possibilities to overestimate or underestimate our findings. Second, for the connectivity metrics computation, all villages located in the same watershed were assumed to be connected. However, is plausible that human travel preferences might involve avoiding certain villages within the same watershed or traveling to villages in different watersheds. Future studies should consider more complex network structures to address these dynamics. While collapsing data for the entire study period (2011–2018) simplifies our understanding of the relative contribution of spatial units containing nodes/villages on malaria incidence, it is important to recognize the loss of temporal granularity. During epidemics, the centrality value of a node can fluctuate significantly, reflecting shifts in malaria transmission dynamics and being influenced by migration patterns and seasonal variations [62]. Finally, previous studies in the Peruvian Amazon [65–67] reported a high number of sub-clinical infections that are not recorded by the MoH during routine data

collection. The findings of this study are relevant only for clinical cases, and caution is suggested when interpreting these results for asymptomatic cases, which can contribute to the maintenance of parasite transmission. Therefore, incorporation of active-case surveillance data will be crucial to address this limitation in future studies.

## 5. Conclusion

This study exploited detailed malaria incidence data at the village level to test the influence of a comprehensive set of connectivity metrics. The data in this study show that in the Loreto department of Peru, villages and cities with high connectivity consistently have higher malaria incidence. When stratified by coverage of PAMAFRO control activities, the areas where malaria transmission was the highest are the areas where this difference in malaria incidence is most pronounced. These findings challenge prior research that emphasized the importance of HPM being greater in low-transmission and close-to-elimination settings rather than in moderate- and high-transmission settings. The evidence outlined in this study can be used to tailor malaria control strategies in rural areas by prioritizing influential connected neighborhoods instead of single villages.

## Supporting information

**S1 Methods. Network analysis–data cleaning.**
(DOCX)

**S2 Methods. Description of centrality metrics.**
(DOCX)

**S1 Table. Descriptive centrality by watershed.**
(DOCX)

**S1 Fig. Data flowchart of the analytical dataset.**
(DOCX)

**S2 Fig. Overall distribution of distance and travel time between dyad villages in the Loreto department in the Peruvian Amazon.**
(DOCX)

**S3 Fig. Distribution of distance and travel time between dyad villages in each watershed in the Loreto department in the Peruvian Amazon.**
(DOCX)

**S4 Fig. Consensus graph of the network of villages by watersheds in the Loreto department in the Peruvian Amazon.**
(DOCX)

**S5 Fig. Correlation of centrality metrics of villages in the Loreto department in the Peruvian Amazon.**
(DOCX)

**S6 Fig. Distribution of Total Annual Parasite Index (API) per 1000 population (2011–2018) across high and low centrality villages stratified by levels of PAMAFRO intervention coverage and overall distributions in the Loreto department in the Peruvian Amazon.**
(DOCX)

## Acknowledgments

We thank the Ministry of Agriculture, Ministry of Health, and the Regional Directorate of Health of Loreto for providing such useful data to researchers. We thank Maren Hale for her help editing the manuscript.

## Author Contributions

**Conceptualization:** Gabriel Carrasco-Escobar, Alejandro Llanos-Cuentas, Tarik Benmarhnia.

**Data curation:** Diego Villa, Antony Barja.

**Formal analysis:** Gabriel Carrasco-Escobar, Diego Villa, Antony Barja.

**Funding acquisition:** Gabriel Carrasco-Escobar, Alejandro Llanos-Cuentas, Tarik Benmarhnia.

**Investigation:** Gabriel Carrasco-Escobar, Alejandro Llanos-Cuentas, Tarik Benmarhnia.

**Methodology:** Gabriel Carrasco-Escobar, Alejandro Llanos-Cuentas, Tarik Benmarhnia.

**Supervision:** Rachel Lowe, Tarik Benmarhnia.

**Visualization:** Gabriel Carrasco-Escobar, Diego Villa, Antony Barja.

**Writing – original draft:** Gabriel Carrasco-Escobar.

**Writing – review & editing:** Gabriel Carrasco-Escobar, Diego Villa, Antony Barja, Rachel Lowe, Alejandro Llanos-Cuentas, Tarik Benmarhnia.

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
