## [Decision Letter · Decision Letter 0]

3 Jul 2024

Dear Mr. Carrasco-Escobar,

Thank you very much for submitting your manuscript "The role of connectivity on malaria dynamics across areas with contrasting control coverage in the Peruvian Amazon" for consideration at PLOS Neglected Tropical Diseases. As with all papers reviewed by the journal, your manuscript was reviewed by members of the editorial board and by several independent reviewers. The reviewers appreciated the attention to an important topic. Based on the reviews, we are likely to accept this manuscript for publication, providing that you modify the manuscript according to the review recommendations. 

Sincerely,

Claudia Ida Brodskyn

Section Editor

Claudia Brodskyn

Section Editor

Reviewer's Responses to Questions

**Key Review Criteria Required for Acceptance?**

**Methods**

-Are the objectives of the study clearly articulated with a clear testable hypothesis stated?

-Is the study design appropriate to address the stated objectives?

-Is the population clearly described and appropriate for the hypothesis being tested?

-Is the sample size sufficient to ensure adequate power to address the hypothesis being tested?

-Were correct statistical analysis used to support conclusions?

-Are there concerns about ethical or regulatory requirements being met?

Reviewer #1: 1. Yes, The objectives of the study are not clearly stated but the authors gave some hints for the reader to guess. Thus, this remain a key issues the authors have to address in the revised version

2. Yes the study design is appropriate even though more clarity could be brought namely why the choice of the centrality metric used, what are the assumption made while using Euclidian distance

3. Yes, the population clearly described and appropriate for the hypothesis tested

4. Yes, the sample size sufficient to ensure adequate power to address the hypothesis tested

5. Yes, the statistical analysis used were correct to support conclusions

6. Yes, concerns about ethical or regulatory requirements being are met and the study is a nonhuman subject

Reviewer #2: As the authors state this is an observational ecological study aimed at testing the relationship between connectivity metrics and

the malaria incidence in the Loreto department of Peru, the administrative region contributing the majority of cases in the country.

**Results**

-Does the analysis presented match the analysis plan?

-Are the results clearly and completely presented?

-Are the figures (Tables, Images) of sufficient quality for clarity?

Reviewer #1: 1. Yes, the analysis presented match the analysis plan

2. Yes, the results are clearly and completely presented

3. Yes, the figures (Tables, Images) are of sufficient quality for clarity

Reviewer #2: The analysis presented -description of the data set describing the characteristics of villages, the choosing and estimation of the different centrality types, and the analysis of the relationship of centrality and malaria cases- does match the analysis plan and is clearly presented.The figures are of good quality.

**Conclusions**

-Are the conclusions supported by the data presented?

-Are the limitations of analysis clearly described?

-Do the authors discuss how these data can be helpful to advance our understanding of the topic under study?

-Is public health relevance addressed?

Reviewer #1: 1. Yes, the conclusions are supported by the data presented

2. Yes, the limitations of analysis are clearly described

3. Yes, the authors discuss how these data can be helpful to advance our understanding of the topic under study

4. Yes, the public health relevance of the study is addressed

Reviewer #2: Conclusions are supported by the data and the limitations are well described as well as its relevance in public health.

**Editorial and Data Presentation Modifications?**

Reviewer #1: This manuscript offers valuable insights into the connection between village connectivity and malaria transmission in the Peruvian Amazon. However, some editorial and data presentation modifications could enhance clarity and strengthen the overall impact:

Editorial Suggestions:

The results section doesn't directly address the research question (likely about the relationship between connectivity metrics and malaria incidence). While it describes various aspects of the data (e.g., baseline characteristics, centrality estimates), it doesn't show how these relate to malaria incidence. Start the results section by summarizing the main findings related to the research question and refocus the discussion on these findings.

Reviewer #2: Minor suggestions

Figure 4. Define meaning of the colors.

As noted below the paper would benefit if discussing what was the impact of PAMAFRO in Loreto given the data presented in figure 6.

A more detailed discussion on the meaning and impact of data collapsing and the impact of metrics changing in timevwould improve the manuscript.

**Summary and General Comments**

Reviewer #1: This study offers a significant contribution by investigating how village connectivity in the Peruvian Amazon impacts malaria transmission. The authors introduce a novel approach utilizing centrality metrics derived from travel time and population data. This method reveals a strong correlation, with villages boasting higher connectivity experiencing significantly higher malaria incidence. This approach presents a valuable tool for informing targeted malaria control strategies that consider network dynamics and potentially accelerate elimination efforts.

However, to further strengthen the report and enhance reader comprehension, some limitations require clarification:

Introduction

1. What is the specific aim of investigating human population mobility and its connection to malaria transmission?

2. How will understanding the role of human mobility inform malaria control efforts and potentially lead to elimination?

Methods

1. While the formulas for Euclidean distance are provided, it would be helpful to explain any assumptions made or limitations of using Euclidean distance instead of actual road network distances.

2. The section cites employing speed restrictions for different land cover categories, however it does not explain how the accuracy of these estimates was verified.

3. Are they any specific reasons why the authors used the six measure of centrality mentioned?

Discussion

1. The authors highlighted various limitations in the paragraph addressing the study limitations, but did not explain how these may affect the findings.

Reviewer #2: This is an important and relevant paper in public health introducing a large-scale network analysis of malaria transmission in Perú, using methods that greatly improve our understanding of malaria transmission and its relationship with human mobility and the connectivity of human settlements. It provides an analytical framework for rational decisions on the way malaria control strategies are conducted and as the authors emphasize may shift control strategies from targeting particular locations towards strategies based on spatial units of human settlement connectivity.

However, in my opinion, and probably not strictly necessary, the discussion would highly benefit if a few points were addressed. 1) What was the long-term impact of PAMAFRO on malaria transmission in Loreto and which would be a sound control policy? This is an important question to address given the results presented. 2) Data were collapsed for the entire study period (2011-2018) and if this seems an appropriate way to simplify our understanding of the relative contribution of spatial units containing nodes/villages on malaria incidence, at the micro-epidemiological level information is lost on how the contribution of a node varies in time (e.g. during epidemics). In this sense, the centrality value for a node may change in time (e.g. Knudson et al, 2020; https://doi.org/10.1038/s41598-020-60676-1); 3) Which is the impact on malaria incidence of nodes with different centrality values in the network? and 4) how to evaluate directed graphs with different indegree and outdegree values for a node? These questions may help produce models of malaria transmission dynamics that in turn may provide information for targeted control interventions.

PLOS authors have the option to publish the peer review history of their article (what does this mean?). If published, this will include your full peer review and any attached files.

Reviewer #1: No

Reviewer #2: No

Figure Files:

Data Requirements:

Reproducibility:

References

---

## [Editor Report · Decision Letter 1]

20 Sep 2024

Dear Mr. Carrasco-Escobar,

We are pleased to inform you that your manuscript 'The role of connectivity on malaria dynamics across areas with contrasting control coverage in the Peruvian Amazon' has been provisionally accepted for publication in PLOS Neglected Tropical Diseases.

Best regards,

Claudia Ida Brodskyn

Section Editor

Claudia Brodskyn

Section Editor

The authors answered all questions raised by the referees and the manuscript is ready for publication.

---

## [Editor Report · Acceptance letter]

17 Oct 2024

Dear Mr. Carrasco-Escobar,

We are delighted to inform you that your manuscript, "The role of connectivity on malaria dynamics across areas with contrasting control coverage in the Peruvian Amazon," has been formally accepted for publication in PLOS Neglected Tropical Diseases.

Best regards,

Shaden Kamhawi

co-Editor-in-Chief

Paul Brindley

co-Editor-in-Chief
